# Placental Immune Responses to Viruses: Molecular and Histo-Pathologic Perspectives

**DOI:** 10.3390/ijms22062921

**Published:** 2021-03-13

**Authors:** Kavita Narang, Elizabeth H. Cheek, Elizabeth Ann L. Enninga, Regan N. Theiler

**Affiliations:** 1Division of Maternal Fetal Medicine, Department of Obstetrics and Gynecology, Mayo Clinic College of Medicine, Rochester, MN 55905, USA; Narang.kavita@mayo.edu; 2Department of Medicine and Pathology, Mayo Clinic College of Medicine, Rochester, MN 55905, USA; Cheek.Elizabthe@mayo.edu; 3Departments of Immunology, Obstetrics and Gynecology, Mayo Clinic College of Medicine, Rochester, MN 55905, USA; Enninga.ElizabethAnn@mayo.edu; 4Division of Obstetrics, Department of Obstetrics and Gynecology, Mayo Clinic College of Medicine, Rochester, MN 55905, USA

**Keywords:** placenta, immune response, viral infection, congenital, SARS-CoV-2

## Abstract

As most recently demonstrated by the SARS-CoV-2 pandemic, congenital and perinatal infections are of significant concern to the pregnant population as compared to the general population. These outcomes can range from no apparent impact all the way to spontaneous abortion or fetal infection with long term developmental consequences. While some pathogens have developed mechanisms to cross the placenta and directly infect the fetus, other pathogens lead to an upregulation in maternal or placental inflammation that can indirectly cause harm. The placenta is a temporary, yet critical organ that serves multiple important functions during gestation including facilitation of fetal nutrition, oxygenation, and prevention of fetal infection in utero. Here, we review trophoblast cell immunology and the molecular mechanisms utilized to protect the fetus from infection. Lastly, we discuss consequences in the placenta when these protections fail and the histopathologic result following infection.

## 1. Introduction

Infections account for approximately 2 to 3% of all congenital anomalies [1]. The placenta is a dynamic organ which is necessary for creating a protective barrier that keeps pathogens out. The complex interplay of the physiologic pathways within the placenta, it’s multifaceted role in both prevention of fetal rejection by the maternal immune system and protection from transmission of infections to the fetus is worth investigating, especially as risks of pandemics increase. Maternal immune responses that occur at the maternal-fetal interface are critical for the acquisition or protection from congenital or perinatal infections.

In this review, the authors provide an overview of the cellular immunopathogenesis of the placental response towards invading infections. We discuss the molecular pathways involved and corresponding placental histopathologies with a specific focus on viral infections.

## 2. Role of the Placenta in Pregnancy Related Immune Responses

### 2.1. Mechanisms of Maternal-Fetal Tolerance

It was Peter Medawar in 1953 who first suggested that a pregnant woman’s immune system becomes inactive to tolerate a semi-allogenic fetus [2]. However, new research has uncovered that adaptation, not suppression, of the maternal immune system occurs during pregnancy. Maternal immune responses are dynamic, with the first and third trimester requiring a pro-inflammatory state for implantation and parturition to occur [3,4]. These responses are tightly regulated by signals originating from trophoblast cells of the placenta. There are three types of trophoblast cells which originate from the trophectoderm layer of the blastocyst: extravillous trophoblasts (EVTs), cytotrophoblasts (CTBs) and syncytiotrophoblasts (SYNs). EVTs invade into the maternal decidua and myometrium, promoting spiral artery remodeling which allows for nutrient transport to the fetus [5]. CTBs are mononuclear cells which make up the placental villi and proliferate in response to human chorionic gonadotrophin (hCG) [6]. Lastly, CTBs can fuse to from SYNs, the main barrier between the maternal immune system and the fetal placenta, which are critical for modulating maternal immunity and protecting the fetus from pathogens [7]. Specific mechanisms that trophoblast cells utilize to modulate maternal immunity will be further described in the next three sections.

### 2.2. Innate Immune Responses by Trophoblast Cells

The first line of control against a foreign cell or pathogen is the innate immune system. This response is non-specific and is driven by placental architecture and recognition of unique molecular patterns not typically seen in eukaryotes. The structure of the placenta makes up the strongest protective barrier through the SYNs, which form large, multi-nucleated cells that lack cell junctions commonly used by pathogens to get into tissues. Additionally, SYN cells have a dense cytoskeletal system which is highly effective at preventing transmission of cells and pathogens from the mother to the fetus, as demonstrated by *Listeria monocytogenes* infection [8]. Next, trophoblast cells can recognize foreign cells and pathogens through Toll-like receptors (TLR) and Retinoic acid-inducible gene I (RIG-I) receptors. TLRs1-10 are expressed on the placenta and recognize viruses (TLRs 3, 7 and 8) bacteria (TLRs 2, 9), and endogenous signals that indicate cell stress (TLR4) [9]. It has been demonstrated that TLR7/8 are increased in trophoblast cells exposed to hepatitis B virus (HBV), preventing maternal-fetal transmission; however, when downstream signaling through MyD88 is blocked, HBV can translocate across the placenta [10]. RIG-I, a cytoplasmic protein, is also expressed in trophoblasts and can recognize both RNA and DNA viruses leading to a conformational change, oligomerization with mitochondrial antiviral signaling proteins (MAVS) and activates transcription of NF-kB [11,12,13]. Others have demonstrated that RIG-I levels are increased upon *Cytomegalovirus (CMV*) and *Herpes simplex virus 1 (HSV-1)* infection [14,15]. Signaling though TLR and RIG-I leads to the production of type I interferons (IFN-α and IFN-β) which induce host anti-viral activity through the activation of both innate and adaptive immune responses.

### 2.3. Cell Mediated Immunologic Responses of Trophoblast Cells

Adaptive immunity is a targeted response to a foreign cell or pathogen and takes a much longer time to generate an effective response compared to the innate response; however, the adaptive immune system generates an antigen-specific memory response so if that same antigen enters the body again it will quickly be destroyed. One component of adaptive immunity is cell mediated immunity which depends on phagocytes presenting foreign antigen to lymphocytes through major histocompatibility complex (MHC) class I and class II. As a fetus gets half of its genetic material from the father, presentation of paternal/fetal antigen would normally induce a specific response against the antigens of future offspring. Therefore, instead of expressing traditional MHC receptors, EVTs express non-classical MHC, including human leukocyte antigen (HLA)-G and -E [16,17], which are capable of processing and presenting foreign antigens through the TAP signaling complex [18] but instead maintain fetal-maternal tolerance [19]. Natural Killer (NK) cells make up a significant percentage of the leukocytes at the maternal-fetal interface; however, unlike peripheral NK cells (pNK) which have a cytotoxic phenotype, decidual NK cells (dNK) take on a tolerogenic phenotype (Figure 1A) [20]. dNK cells acquire this immunomodulatory function through signaling between natural killer group 2 receptor (NKG2) [21] and HLA-E as well as immunoglobulin-like transcript 2 receptor (ILT-2) with HLA-G [22]. EVTs further influence tolerance by blunting antigen-specific T cell cytotoxicity and promoting the differentiation of naive T cells towards a regulatory phenotype [19,23]. Additionally, EVTs do express class I HLA-C, which can influence NK cell phenotypes by restricting anti-fetal responses through killer cell Ig-like receptors (KIR) [24]. Trophoblast cells also express costimulatory receptors such as program death ligand I (PD-L1) and galectin-9, which interact with program death 1 (PD-1) and T cell immunoglobulin mucin 3 (Tim-3), respectively, to blunt T or NK cell activation [25,26,27]. Together, these cellular mechanisms ensure tolerance to the allogeneic fetus locally while systemically maintaining maternal defense against pathogens.

### 2.4. Humoral Mediated Immunologic Response in the Placenta

Another component of adaptive immunity is the humoral response, which is driven by the production of antibodies from activated B cells called plasma cells. These antibodies will recognize and bind to foreign pathogens, leading to their elimination through complement activation, neutralization, and enhancement of phagocytosis (opsonin). In the case of intracellular pathogens, antibodies binding to critical surface proteins, including virus spike proteins, may directly block cell entry. During pregnancy, passive immunity is acquired by the fetus from the mother through the transplacental passage of maternal IgG antibodies [28]. Transfer of these protective antibodies is mediated through Fc-receptors, specifically FcRn, on the SYN cells, where IgG within endosomes travel through the interstitial space be released into the fetal circulation [29,30,31]. Interestingly, maternal-fetal transfer of IgG is decreased during maternal infection with HIV and malaria [32,33]. IgG can also activate the complement system, leading to enhanced phagocytosis and protection from infection. Trophoblast cells express complement proteins, including C1q, C4d, CD46, CD55 and CD59 [34,35]. While complement activity is tightly regulated to prevent placental injury, in the case of infection the process of targeting the pathogen leads to inflammation, endothelial activation and coagulopathy which can harm the fetus [36]. Together, these regulatory processes in the placenta create a formidable blockade that protects the fetus from infection while facilitating maternal tolerance of the fetal hemiallograft; however, some pathogens have found unique mechanisms to exploit these placental safeguards which we cover in the next few sections.

## 3. Mechanisms of Viral Infection of the Placenta

Viral pathogens that reach the placenta do so via hematogenous spread, requiring that significant maternal viremia be present for placental infection to occur. The requirement for significant maternal viremia explains why some localized secondary infections, including shingles or recurrent HSV infections are unlikely to cause placental and fetal infections. As the SYN barrier directly contacts maternal blood as it bathes terminal villi, viral infection of the SYN itself is a usual precursor to transmission of infection to the fetal blood and downstream organs. While many viruses, including HIV, CMV, and SARS-CoV-2 directly infect the placenta by binding to viral receptors on the maternal aspect of SYNs, others may use antibody- dependent enhancement (ADE) to cross the SYN barrier. Viral replication in any cell is imperfect, resulting in newly synthesized viral proteins and nucleic acids being present throughout the infected cell and presented on the cell surface. Newly translated viral peptides are also processed for cell-surface antigen presentation in the context of MHC class I and II, along with the inhibitory HLA-E and HLA-G on trophoblasts. Members of the HSV family, and CMV, employ complex strategies for immune evasion. During infection, CMV engages mechanisms to evade the innate, cell-mediated, and humoral arms of the host immune response. In the setting of placental infection, evasive strategies by CMV, Zika virus, and other placental pathogens are synergistic with pre-existing adaptations of the maternal and fetal immune responses in the placenta, allowing for robust viral replication in this immunologically protected organ.

Virus-encoded mechanisms for evasion frequently target components of the type I interferon pathway. Congenital pathogens known to down-regulate this pathway include: Zika Virus, HIV, CMV, SARS-CoV-2, and others. Normally, infection with these viruses induces transcription of pro-inflammatory cytokines and type I interferons, resulting in the downstream establishment of an antiviral state in the infected cell by inducing transcription of hundreds of interferon-stimulated genes (ISGs). The ability of the placenta to mount an innate immune response is more robust at later times during gestation, and viruses may thus be more successful at productively infecting the placenta during early gestation [37,38]. CMV, Zika, and many other viruses employ multiple mechanisms to exploit the vulnerability of the placenta. Zika virus NS5 inhibits type I interferon receptor signaling and blocks induction of type I interferon via the RIG-I pathway [39,40,41]. (Figure 2) Like Zika and other flaviviruses, CMV uses multiple strategies to impair the innate host response to infection. CMV IE1, which is produced immediately upon infection, binds to and sequesters STAT2, preventing upregulation of ISGs [42].

Further downstream, CMV IE86 inhibits production of the pro-inflammatory cytokine IL-1β [43]. Other CMV gene products involved in inhibition of the innate response include UL44, IE2, and UL94 [42,44,45]. Both CMV and Zika virus, model congenital pathogens, target the host interferon response while causing robust infections during early gestation when innate immunity in the placenta is less effective.

Evasion of cell-mediated immunity is also known to occur during CMV and other herpesvirus infection, and viral inhibitory mechanisms are additive with changes already present in SYNs. In addition to the inherent SYN resistance to NK cell recognition conferred by cell-surface display of HLA-E, CMV encodes additional inhibitory molecules targeting antigen presentation and NK cell recognition. Mechanisms include downregulation of NK cell activation receptors and classical MHC I complexes, upregulation of HLA-E production, and expression of a virus-encoded inhibitory MHC decoy (UL18) [46]. Paradoxically, Zika virus upregulates expression of MHC class I in many cell types (Figure 1B). This is thought to decrease NK cell recognition and killing of infected cells but does not have the additional T cell inhibitory effects of HLA-E expression as seen with CMV infection [47].

Viral pathogens have fewer known mechanisms of evading humoral immunity. Interestingly, the unique maternal-fetal IgG transport pathway of the trophoblast makes the placenta and fetus susceptible to some viral infections. Antibody-dependent enhancement (ADE) allows for low-affinity virus-specific IgG to facilitate transport of Zika virus across the cell membrane via FcR-mediated endocytosis. ADE is thought to be one mechanism by which Zika virus efficiently infects the placenta in women with previous Dengue virus infections (Figure 3) [39]. Both HIV and CMV have also been shown to use ADE to traverse the trophoblast during early infection when low-avidity IgG antibodies predominate [48,49,50].

## 4. Clinical Pathophysiology of Specific Infections in Pregnancy

Several physiologic changes occur in pregnancy including significant adaptations in the maternal immune system to prevent rejection of an allogenic fetus. Consequently, this may increase the risk of certain maternal infections which may subsequently impact the fetus. It has been shown that EVT cells are more permissive to the transmission of infection compared to the SYN. Therefore, the window of transition from first trimester to second trimester is a particularly vulnerable time for infections due to decreased SYNs and increased EVT. This vulnerability starts decreasing from second trimester into third trimester due to the increased presence of SYN cells in the placenta.

Though there are numerous disease-causing infections, common viruses that result in significant congenital infections are reviewed in detail below. In addition to these infections, the SARS-CoV-2 virus which has resulted in the COVID-19 pandemic, is also of significant importance and will be discussed here.

### 4.1. Cytomegalovirus

The cytomegalovirus (CMV) is a double stranded DNA virus belonging to the class of herpesviruses. The incidence varies based on geographic distribution and socioeconomic status; with rates of 0.7% in developed countries [51] and as high as 1–5% in developing countries [52]. It is transmitted by sexual contact or direct contact with infected body fluids including blood, urine and saliva and be classified as primary (infection in seronegative individual) or secondary (reactivation of infection) [53]. CMV is the leading cause of congenital viral infection, occurring in 0.2 to 2.2% of neonates and resulting in deafness and neurodevelopmental delay [51]. Congenital infection occurs through vertical transmission across the placenta, fetal exposure to contaminated genital tract secretions at the time of delivery or through breastmilk. As seen with other congenital infections, the risk of transmission increases with advancing gestational age, having reported rates of 5.2%, 16.4%, 36.5%, 40.1% and 65% in the preconception, periconceptional, first trimester, second trimester and third trimester, respectively [54]. However, the risk of disease severity is higher when infection occurs at earlier gestational ages and in the setting of primary infection [54]. Sonographic findings suggestive of congenital CMV include abdominal and liver calcifications, hepatosplenomegaly, echogenic bowel or kidneys, ascites, cerebral ventriculomegaly, microcephaly, hydrops and fetal growth restriction [55] If maternal serologic findings and sonographic findings are concerning for congenital CMV; amniocentesis for amniotic fluid CMV PCR can be performed for diagnostic testing. However, the detection of CMV in the amniotic fluid does not correlate with disease severity [56].

CMV has been identified in villous endothelial cells, epithelial cells of endometrial glands and endothelial cells of lymphatic vessels within decidua, EVTs, CTBs and Hofbauer cells [57,58]. In vitro studies of first trimester chorionic villi and isolated CTBs exposed to CMV identified transplacental transmission of the virus to occur via one of two pathways: 1) across SYNs via ADE with subsequent infection of CTBs, or 2) via direct infection of invasive CTBs in the uterine [53]. Inflammatory pathology from the anti-CMV placental immune response has been described in several studies. Following CMV infection of CTB and SYN, tumor necrosis factor (TNF)-α is secreted, causing apoptosis of uninfected cells in a paracrine fashion [59,60,61]. This inflammatory response at the maternal-fetal interface has detrimental effects on the placental structure and function that are separate from the effects of the infection itself.

Despite the varying risk of transmission throughout gestation, the data indicates the placenta also serves as a protective barrier during maternal infection [62]. As placental defense mechanisms are studied butremain largely enigmatic for many organisms, it is proposed that viral interactions between cells of the uterine and basal plate (vasculature, leukocytes, and CTBs) impact the severity and outcome of CMV infection [61]. Co-receptors-like integrin α5β3, α2β1 and α6β1, are necessary for internalization of CMV particles into fibroblasts and have been identified on CTBs. Studies that block these integrins prevent CMV entry into the placenta [63]. Lastly, the “double-hit-hypothesis” proposes that two concurrent infections weaken the defense, causing a more severe infection, as can be seen with CMV and with coinciding bacterial infections [64] and co-infection with HIV [65,66].

Microscopic manifestations of placental CMV infections vary with gestational age and infection progression (Figure 4). Studies have noted delayed villous maturation, villitis, cytomegalic cells, necrosis, and calcifications in term placentas when congenital CMV was diagnosed at mid-gestation [67]. Others have shown villous necrosis, but with infected endothelial cells identified in villous stroma not showing pathologic changes [62,68]. The viral cytopathic effect gives rise to the classic “owl-eye” inclusions, which are seen in only approximately 10% of cases [53,62,68]. As a result, IHC studies are an invaluable asset, highlighting insidious viral inclusions and/or viral proteins within the placenta. Infiltration of villous capillaries and villous stroma impacts villous development, characterized by delayed villous maturation. Longstanding infections lead to vascular sclerosis and obliteration, chorionic vessel thrombi, chronic lymphoplasmacytic villitis, and multifocal calcifications [68,69].

### 4.2. Hepatitis Viruses

Viral hepatitis is a common infection which can adversely affect pregnant women and their fetuses. Known viral hepatitis include Hepatitis A, B, C, D and E. Vaccinations are currently available for Hepatitis A and B; however, a large portion of women worldwide remain unvaccinated. Vertical transmission and clinical consequences are highest in patients with acute HBV infection. The HBV virus contains three principal antigens; namely Hepatitis B surface antigen (HBsAg) which is present on the viral surface, Hepatitis B core antigen (HbcAg) which is at the center of the viral particle, and Hepatitis Be antigen (HbeAg) which is coded by the same gene that codes for the core antigen.

The risk of HBV vertical transmission is approximately 10–20% in women with positive HBsAg and 90% in women with both HbsAg and HbeAg [70]. Perinatally acquired HBV infection confers 85–95% risk of chronic infection in the infant [71]. The frequency of vertical transmission also increases with advancing gestational age; transmission rates are 10% in the first trimester and up to 90% in the third trimester [70]. Transmission can occur prenatally or at birth. Due to this high risk of vertical transmission, routine screening of pregnant women for HbsAg at the first prenatal visit is recommended. Administration of Hepatitis B immunoglobulin (HBIG) can provide an immune barrier by limiting transcytosis of HBV and reduces the risk of vertical transmission of HBV in pregnancy. The HBIG antibodies can passively diffuse across the placenta at the materno-fetal interface; this process occurs at its maximum during the third trimester, which is the recommended window for administration of HBIG, with a dose of 100 IU to 200IU [72] In addition, HBIG should also be administered to neonates in the immediate postnatal period, within 12 h of birth [72]. Studies have reported varying mechanisms of transplacental viral transfer including a breach in the placental barrier during uterine contractions resulting in leakage of HbeAg [73], cell to cell transfer of virus from the maternal interface of an infected placenta [74,75]. In vitro studies also suggest that HBV results in intrauterine infection via infection of the placental trophoblasts; as seen by detection of HBV DNA in trophoblastic cells incubated with HBV positive serum [76].

Like HBV, HCV can also be vertically transmitted to the fetus prenatally or during delivery. The risk of transplacental transmission of HCV during chronic maternal infection is approximately 2–8% [77,78]. Studies propose that transmission occurs by transcytosis, transcellular vesicular transport of the virus, receptor mediated entry and/or infection of trophoblasts [79]. Placental infection with HCV results in damage of the placental barrier through enhanced placental NK T cell and γδ T-cell cytotoxicity while reducing expression of NK cell activation markers CD69, NKp44 and TRAIL creating a pathway for HCV transmission [80]. There are currently no vaccinations for HCV and no preventive strategies to reduce the risk of vertical transmission.

Reported histopathologic changes that occur following HBV infection of the placenta include stromal fibrosis, syncytial knotting, fibrinoid deposition, fibrinoid necrosis, villous capillary congestion and proliferation [81]. Xu et al. studied a cohort of 101 full-term placentas from HBsAg-positive pregnant women and reported HBV infection rates decreased in cell layers from the maternal side to the fetal surface. Large deposits of bilirubin can be seen in Hofbauer cells (fetal macrophages), trophoblasts, and macrophages of the fetal and extraplacental membranes in active HBV infections [82]. As HBV nucleocapsid contains two serologically distinct antigens (core and envelope), with a surrounding outer envelope surface antigen, the IHC stains both cytoplasm and membranes [76].

### 4.3. Varicella Zoster Virus

Varicella Zoster Virus (VZV), a Herpesvirus, causes varicella zoster (chicken pox) at time of primary infection and herpes zoster (shingles) at time of recurrence or reactivation of the latent herpes virus. Since the introduction of childhood VZV vaccination in 1995, the incidence of maternal varicella has decreased significantly [83]. The risk of congenital varicella syndrome is thus exceedingly rare given the low incidence of maternal disease. A large prospective study found the incidence of congenital varicella syndrome to be 0.4% if infection occurred before 12 weeks gestation, and 2% if infection occurred between 13 and 20 weeks of gestation [84]. Only 9 cases have been reported beyond 20 weeks of gestation [85]. The virus is transmitted from person to person via respiratory droplets to the nasal, oral or conjunctival mucosa, and via direct contact with vesicular fluids that contain the virus. Therefore, during pregnancy, mother to fetus transmission is possible in utero, perinatally, or postnatally.

Once the mother has a primary VZV infection, the virus replicates in various organs and lymph nodes, resulting in maternal viremia that may infect the placenta. The VZV DNA can be detected in multiple fetal organs, most notably in neural tissue, and infection contributes to limb hypoplasia, neurologic deficiencies (damage to autonomic nervous system) [84] and intrauterine encephalitis observed in infants with congenital varicella syndrome [86].

The primary mechanism of VZV transfer across the placenta remains unclear; however, it is postulated that T-cells infected with varicella virus bathe the decidua basalis, where the virus replicates and spreads into the intervillous space [87]. Both CD4 and CD8 T-cells are reprogrammed following VZV infection, rendering them more capable of crossing into the intervillous space [88]. This builds on previous studies showing that activated CD4 memory T-cells are highly susceptible to varicella infection [89]. A more recent study utilizing human tonsil tissue reported that the virus has capabilities to remodel naïve and memory T-cells, producing cell surface markers that favor tissue trafficking to the skin [88]. To counteract the immune system, VZV blocks JAK kinases from phosphorylating STAT1, thereby preventing the production and release of antiviral IFN-α and increasing VZV titers [90].

Reports vary on the histological features of VZV placental infection. Some studies indicate that regardless of a maternal history of varicella during pregnancy and time elapsed between maternal infection and birth, no viral-associated microscopic features are appreciated, perhaps indicating that VZV is transmitted to the fetus via the placenta, without apparent viral replication within the placenta [91,92]. Qureshi et al. [91] also reported nonspecific features, such as diffuse basal chronic villitis, mixed inflammatory infiltrate (lymphocytes, histiocytes, and multinucleated giant cells [Figure 5A]) and occasional nuclear clearing with faint eosinophilic center. Others have reported acute inflammation and diffuse or necrotizing chronic villitis with foci of granulomatous inflammation [93,94]. Immunohistochemistry images of the placenta usually reveal membranous and cytoplasmic staining, this is secondary to the VZV immunohistochemical stain which utilizes a mix of antibodies that highlight several glycoproteins, the nucleocapsid protein and the immediate early protein of the virus.

### 4.4. Parvovirus B19 Virus

Parvovirus B19 is a single-stranded DNA virus that results in non-immune hydrops in fetuses, erythema infectiosum in children, and athropathies/myocarditis in adults [95]. The virus spreads mainly through respiratory droplets, infected blood products and by vertical transmission from mother to fetus [96]. Reported transmission rates include 50% with individuals residing with infected persons and approximately 20–30% for susceptible daycare workers and teachers exposed to infected students [97]. Within North America, approximately 65% of pregnant women show serologic evidence of prior infection, with the incidence of infection during pregnancy between 1–2%, increasing up to 10% during an epidemic [98]. Most women remain asymptomatic during an infection, although the transplacental transmission rate is 17–51% in acutely infected parvovirus B19 patients [99,100,101,102]. Maternal serologic screening is not routinely performed unless there are suspicious clinical findings or exposure. Management during pregnancy is dependent upon the severity of fetal anemia and gestational age [95]. For severe fetal anemia, intrauterine fetal transfusion of red cells is advised, with possible platelet administration to abate fetal thrombocytopenia. In mild cases of fetal anemia, invasive therapy is not recommended, however, the pregnancy will be followed by serial ultrasound examination and fetal middle cerebral artery (MCA)-peak systolic velocity (PVC) assessment [95]. In utero infection is suspected based on sonographic findings such as non-immune hydrops, placentomegaly, and stillbirth [103]. At 23 weeks’ gestation, the incidence of intrauterine fetal demise peaks, and is linked to the presence of sphingolipid globoside, abundant fetal erythropoiesis causing high viral loads and erythropoietin signaling from low fetal oxygen levels, all which support viral replication [57,100]. The main cellular receptor for Parvovirus B19 is the P blood group antigen globoside, that is strongly expressed on SYN and CTB cells and erythroid lineage cells. It is thought the presence of this receptor along trophoblast is involved in the virus’s ability to transmit across the placenta [104]. P-antigen is abundant in the placenta during the first half of pregnancy [98], increasing the risk of placental infection and transmission to the fetus. Although the cell-mediated immune response to placental parvovirus infection is not well characterized, one study noted a significant increase in interleukin-2 (IL-2) production in placentas from women with parvovirus B19 infections compared to uninfected placentas [104]. They also found that nearly 90% of the parvovirus-infected placentas had either a CD4 T-helper or CD8 cytolytic infiltrate into the villi.

Aside from identifying the eosinophilic circular viral inclusions within erythroid precursors in villous blood vessels, placental histological features also include hydropic chorionic villi, villous immaturity, nucleated red blood cells, and occasional chronic villitis (Figure 5B) [105]. If the inflammatory response is robust enough, placental dysfunction can occur leading to adverse fetal outcomes in the absence of fetal infection [98]. One study reviewed placentas from six women infected with parvovirus B19 during pregnancy and confirmed a mononuclear infiltrate present within the intervillous space and the chorionic villous stroma [106]. IHC is also available to highlight viral inclusion bodies in the nuclei of erythroid precursors, with viral antigen being detected in a cytoplasmic and nuclear pattern [107]

### 4.5. Human Immunodeficiency Virus

Human immunodeficiency virus (HIV) is a globally prevalent infection that is transmitted via sexual contact, exposure to infected blood or through perinatal transmission. Maternal to fetal transmission can occur in utero, during delivery or through breastfeeding. The risk of vertical transmission in utero is approximately 1 to 2% and is proportional to maternal CD4 counts and viral load [57]. American College of Obstetrics and Gynecology (ACOG) currently recommends screening all pregnant women for HIV at entry into prenatal care and at 28 weeks gestation. All pregnant women diagnosed with HIV should be started on antiretroviral therapy, such as Zidovudine. This specific drug is metabolized into its active form in the placenta and inhibits HIV replication specifically in trophoblast cells [108].

In vitro studies have identified that HIV- infected lymphocytes target the trophoblastic cell layer through endocytosis or transcytosis of the virion particle by direct cell-to-cell contact [75,109]. Pathologic studies of the human placenta identified CD4 receptors on trophoblastic cells and terminal villi [110], making them susceptible to infection by HIV. Additionally, there is evidence of HIV genomic material within CTB, SYN and Hofbauer cells [111]. Thus, HIV replication and transmission in the placenta may occur through CD4 positive Hofbauer cells or endothelial tissues [112]. Additionally, HIV uses the CCR5 and CXCR4 chemokine receptors to enter host cells, and these receptors are also expressed on placental tissue, further facilitating transfer of the virus [112]. As pregnancy results in a shift from Th1 to Th2 maternal immune responses, antiviral responses lead to Th1 cytokine production resulting in upregulation of CCR5 and CXCR4 receptors [113]. Consequently, placentas from HIV- transmitting mothers have a predominance of Th1 immunity, whereas those of from non-transmitting mothers sustain a strong predominance of Th2 immunity throughout pregnancy [114].

Regarding placental immune defense, Hofbauer cells act as an important mediator at the feto-maternal interface during ongoing HIV-1 infection. These specialized placental macrophages can isolate antibodies and antibody-virion immune complexes through expression of the following HIV-1 receptors along their surface: CD4, CCR5, CXC45, Fcγ and DC-SIGN [115,116,117]. Johnson et al. report in vitro models by which Hofbauer cells, although a target for HIV-1 infection, reduce mother-to-child transmission of HIV-1 by inducing immunoregulatory cytokines IL-10 and TGF-β1 [117]. Interestingly, these elevated levels of cytokines may indicate a mechanism by which the placenta provides an antiviral response at the maternal-fetal interface [115].

Microscopically, placental changes from an HIV infection have been described to impact villous maturity, characterized by distal villous immaturity, increased villous cellularity, and decreased vascularity [82]. In contrast, others have noted no associated placental lesions specific to HIV, despite in cases in which the virus was detected via cultures or electron microscopy [118]. Through IHC and in situ hybridization, the virus has been detected in villous trophoblasts, endothelial, and stromal cells as early as 8 weeks gestation [82,111,119]. Electron microscopy is an alternative method for identifying the presence of HIV. Using this technique, the virus can be appreciated within the SYN layer, cells of the decidua, umbilical vessels, and less commonly in Hofbauer cells [82,120].

### 4.6. Rubella Virus

Rubella virus is a member of the *Togavirus* family and is transmitted via direct droplet contact from nasopharyngeal secretions. The virus spreads from the respiratory tract, into the blood via infected lymphocytes and alveolar macrophages to local lymph nodes, causing lymphadenopathy [57]. This is followed by dissemination through the body to involve joints, skin, and the placenta, resulting in congenital rubella infection. Since the introduction of childhood Rubella vaccine, the incidence of congenital rubella has significantly decreased to less than 0.1 per 100,000 [121], and is usually seen in parts of the world where access to vaccination is challenging. Though rare, transplacental transfer of rubella can result in congenital rubella infection associated with spontaneous abortion, stillbirth and fetal growth restriction, or congenital rubella syndrome associated with birth defects including cataracts, congenital heart disease, hearing loss and fetal death [122]. These consequences occur secondary to virus- induced inhibition of cell division and cytopathic damage of affected fetal organs and blood vessels [123,124]. The risk of fetal infection with rubella is highest in the first trimester, especially prior to 10 weeks of gestation [125]. ACOG recommends routine screening for rubella immune status in all pregnant women. Those who are non-immune should receive MMR vaccine postpartum for protection against future infections with measles, mumps and rubella [121].

The exact mechanism by which Rubella traffics to the maternal-fetal interface has not been well studied. One hypothesis suggests that chronic infection causes monocytes to disseminate into the intervillous space and/or the lymphatic vessels within the basal plate. Placentas from congenital rubella syndrome were noted to have detectable antigens in CTBs, endothelial cells of villous capillaries, amniotic epithelium and various cells of the basal plate [126,127]. Adamo et al. [127] reported Rubella induces apoptosis in chorionic villous explants from human term placentas but not primary human embryo fibroblast cultures, suggesting the etiology for persistent congenital infection. In their models, they also showed elevated pathogenicity in human cell models compared to Vero monkey cell cultures, as humans are the natural host for Rubella. They postulate the persistence of the congenital infection is due to the intracellular location of the virus and its ability to avoid particular antibodies, the inability to activate apoptosis in parts of the infected cells during embryogenesis, and the slower growth rate limiting the duplication seen in clones of infected cells [127]. Others [128] have proposed the pathogenesis to include necrosis of chorion and endothelial cells, resulting in circulation of these infected cells to fetal organs such as eyes, heart, brain, and ears. Furthermore, Adamo et al. [127] discuss that the immune system may play a role in the pathogenesis, causing interferon and cytokines CCL5 or RANTES to be upregulated in fetal rubella-infected cells, contributing to growth and proliferation disruption of various differentiating cells, leading to congenital defects.

The histopathological findings in the rubella infected placenta include acute and chronic villitis and intervillositis, with necrosis of trophoblast cells and villi, vasculitis, and obliteration of stem villi [69,82]. In addition, there is delayed villous maturation and edema of stem and terminal villi [67]. Involvement of the umbilical cord demonstrates a generalized vasculitis with endothelial necrosis, thought due to the endothelial fixation of the virus [82]. Other reports describe an absence of destructive villitis and that the membranes, umbilical cord and decidua have chronic mononuclear inflammatory infiltrate, with mononuclear cell involvement of the decidua being one of the more common findings in rubella placentas [106]. Rubella viral inclusions appear eosinophilic round, intranuclear or cytoplasmic, and can be appreciated within the cells of decidua, EVTs, endothelial cells and amnion (Figure 5C) [69]

### 4.7. Herpes Simplex Virus

Herpes simplex virus (HSV) is a double-stranded DNA virus classified into HSV-1 (common cause of herpes labialis) and HSV-2 (common cause of herpes genitalis) based on its glycoprotein lipid bilayer envelope G1 and G2, respectively. It is transmitted through direct contact with sores, saliva, or genital secretions. Infection by HSV can be classified into three groups: primary, non- primary first episode, and recurrent infection. Primary infection is diagnosed when HSV-1 or HSV-2 is detected from lesions, without antibodies to any of the viral strains. Non- primary first infection refers to confirmation of either HSV-1 or -2 with antibodies to the other viral strain. Recurrent infection refers to detection of either HSV-1 or 2 and antibodies to the disease-causing viral strain. Approximately 1.6 million new cases of HSV-2 are diagnosed annually in the United States [129], and the incidence of new HSV-1 and HSV-2 infection diagnosed during pregnancy is 2%. Highlighting the insidious and ubiquitous nature of HSV, 60–80% of women that deliver an HSV-infected infant have an asymptomatic genital infection [130,131]. Interestingly, there is a lower risk of transmission to an exposed infant in recurrent HSV-2 lesions compared to new infection (2% vs. 57%) [132,133]. Neonatal HSV infection mainly presents with three clinical manifestations: (a) skin, eyes, and mouth herpes (SEM) disease, usually presenting as a vesicular rash occurring at 10–12 days of life; (b) CNS disease, with (60–70%) or without skin involvement, with clinical manifestations of encephalitis, potentially starting at any time within the first month of life and (c) disseminated disease, involving multiple organs including CNS, lungs, liver, adrenal, skin, eye, and/or mouth, with about 40% of infants never developing a vesicular rash [133].

The transmission of HSV to the neonate usually occurs through direct neonatal contact with infected maternal lesions at the time of delivery. Once infected, most women will develop viral specific antibodies after 12 weeks and this confers some protective effects to peripartum neonatal infection. This is supported by the findings that there is a higher risk of neonatal infection in newborns of mothers with primary or nonprimary first episode HSV-2 occurring near the time of labor and delivery when viral specific antibodies are more likely to be absent [134].

Robb et al. [135] proposed that transplacental HSV-1 infections may arise during transneural migration from perpetually infected dorsal root ganglia to the endometrium, or via direct placental infection through latently infected endometrium that is reactivated during pregnancy. EVTs are susceptible to infection by HSV-1 via entry mediators HveA, HveB, and HveC, which impact placental implantation resulting in dysfunction during early pregnancy and may lead to miscarriage [7]. Additionally, McDonagh et al. evaluated the relationship between pathogenic bacteria and concurrent infections with CMV and HSV from placental and decidual samples. They noted that HSV-1 and -2 infections were more focal within the decidua and that viral DNA was not found in the adjacent SYN cells of the placenta [136]. Attachment of the virus is facilitated by heparan sulfate, with entry being regulated by mediators HveA, HveB, and HveC [7]. The binding to heparan sulfate arises from interaction with virion glycoprotein C and then entry by the entry mediators [137,138]. Interestingly, enzymatic removal of heparan sulfate from cells reduces HSV’s ability to attach and cause infection by approximately 85% [7]. Koi et al. [7] report the placenta provides a defense mechanism against HSV through the SYN layer, limiting vertical transmission by not expressing the viral entry mediators HveA, HveB, and HveC. Koi et al. [7] further demonstrate that antibodies against HveB most effectively prevented HSV-1 entry into EVT cells [7].

Identification of HSV within the placenta is relatively uncommon, with a large study identifying a total of 64 cases between 1963 and 2009, supporting the rarity of intrauterine infection [139]. Histological manifestations of a placental HSV infection can include subacute necrotizing inflammation with stromal cell necrosis involving the umbilical cord and fetal surface/chorionic plate (Figure 5D) [140]. Additional investigation has described chronic chorioamnionitis with degradation of the amniotic lining, chorionic villi that are appropriate for gestational age, delayed villous maturation, and a HSV Cowdry type B intranuclear inclusion within subamniotic tissue [141]. Another study evaluating 64 cases of latent neonatal infection endorsed that although no HSV cytologic changes were found, the HSV-specific antigen was identified in individual cells of the subamnionic chorion, and/or individual cells of the subamniotic tissue, and/or perivascular tissue [135]. Through IHC, staining for HSV types 1 and 2 produce nuclear and granular cytoplasmic staining of the infected cells.

### 4.8. SARS-CoV-2 Virus

The severe acute respiratory syndrome coronavirus 2 (SARS-CoV-2) is an enveloped single-strand positive-sense RNA virus identified in December 2019 and is the etiologic organism of COVID-19 disease, a pandemic declared by the WHO in March of 2020 [145]. Evidence of the impact of SARS-CoV-2 in pregnancy is rapidly evolving and data on vertical transmission is limited. The effects on the fetus discussed in literature have included increased incidence of preterm delivery, higher rates of miscarriage, perinatal death, and intrauterine fetal distress [146].

SARS-CoV-2 enters cells by through binding its spike (S) protein to angiotensin-converting enzyme II (ACE2) receptor, with subsequent endocytosis, genomic replication, and mature virion release via exocytosis [147,148]. Reported main sites of ACE2 expression in the placenta have included the villi (SYN, CTB, endothelium of the villous vasculature, and smooth muscle cells of the primary villi), EVTs and decidual cells, and ACE2 is highly abundant in the early gestational placenta [149]. Additionally, SARS-CoV-2 utilizes S protein for receptor recognition to fuse with cell membranes [148,150]. However, in order to be completely functional, S protein requires proteolytic cleavage by human transmembrane protease serine 2 (TMPRSS2), cathepsin L (CTSL), furin, elastase, factor X or trypsin [148,151,152]. Single cell data of placental tissue from each trimester demonstrated that TMPRSS2 expression is low throughout the placenta, which likely explains why in utero infection has rarely been reported [153]. As SARS-CoV-2 research is ongoing, other proposed receptors for entry have surfaced, including dipeptidyl peptidase 4 (DPP4) and CD147/Basigin (BSG) [154]. Placental histopathology of third trimester placentas from 150 SARS-CoV-2 patients amongst 20 studies found that 2% of neonates and 21% of placental samples tested positive for SARS-CoV-2 [145,149]. SARS-CoV-2 is predominantly observed in the SYN layer with low levels in the villous CTB, indicating that the SYN cells provide a protective barrier against fetal transmission. A study by Kreis et al. evaluated the immune response of the placenta against SARS-CoV-2 and concluded that vertical transmission may occur, but this event is rare, owing to the effective placental innate immune response which induces type III IFN signaling, secretion of chromosome 19 miRNA cluster (C19MC) to restrict viral infections in autocrine and paracrine manners and regulation of the NF-κB pathway [148].

In addition to pathology, transcriptomic profiling using nucleic acid sequencing approaches have also demonstrated utility in understanding SARS-CoV-2 pathogenesis. A retrospective case-based analysis of liveborn neonates infected with SARS-CoV-2 via transplacental transmission, determined using RNA in situ hybridization against SARS-CoV-2 virus, identified SYN necrosis and chronic histiocytic intervillositis in the placenta [155]. Quantitative RT-PCR targeted to SARS-CoV-2 nucleocapside proteins showed that in one case of maternal-fetal transmission, viral titers were highest in the placenta (~10^10^ copies/100mg) followed by the cord blood (>10^5^ copies/1mL); however, the virus was not identified in the fetal heart or lung tissue [156]. Lastly, single cell RNAseq from placental cells of COVID-19 hospitalized mothers demonstrated that although no viral transmission occurred during pregnancy (as defined by a negative SARS-CoV-2 RT-PCR test), there was altered gene expression observed which included increased expression of inflammatory cytokines, innate immune pathways and proteins critical for cytotoxicity as compared to uninfected controls [157]. Combining results from molecular pathways and pathological observations is a necessary step to further our knowledge and prenatal clinical care strategies for maternal SARS-CoV-2 infection and any future outbreaks.

Histopathological findings of SARS-CoV-2 infection in the placenta are met with conflicting opinions (Figure 5E). An insightful literature review by Sharps et al. summarizes 50 studies and discusses features of fetal vascular malperfusion (35.3%), maternal vascular malperfusion (46%), villitis (8.7%), chorioamnionitis (6%) and intervillositis (5.3%) being a trend [145]. Others have reported an increase in perivillous fibrin deposition in varying quantities [158,159,160,161]. In contrast, Hecht et al. reported on a series of 19 placentas, in which similar pathologies were identified; however, they concluded that given identical lesions were present in their control groups, SARS-CoV-2 is not associated with specific placental histopathology [162]. Interestingly, they also add that they suspected to see increased incidences of chronic villitis and chronic intervillositis, as these chronic inflammatory pathologies are more commonly reported in RNA viral placenta infections [162].

## 5. Conclusions

The understanding of placental molecular, immunologic and histopathic pathways and their role in transmission of, or protection from, infection is paramount to the care provided during pregnancy as well as the outcome of the infant. Continued research and focus in this area are critical to defining risks of vertical transmission of emerging infections as well as addressing prevention and treatment of existing infections.

## Figures and Tables

**Figure 1 ijms-22-02921-f001:**
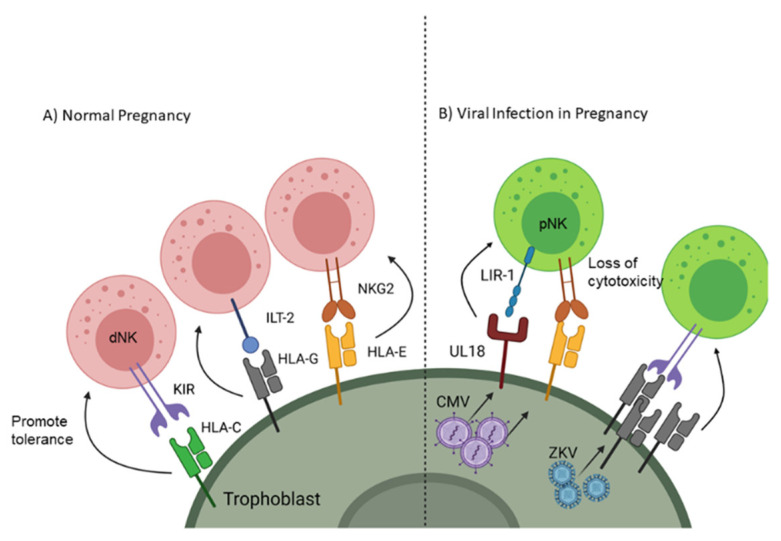
Receptor signaling on trophoblast cells to promote natural killer (NK) cell tolerance to the fetus and viruses. (**A**) Non-classical MHC interactions with NK cells promotes decidual (d)NK phenotype and tolerance. (**B**) Overexpression of non-classical MHC and upregulation of decoy receptors are used by viruses (CMV and Zika virus (ZKV)) to blunt peripheral (p)NK cytotoxicity. Created with BioRender.com. Accessed on 6 February 2021

**Figure 2 ijms-22-02921-f002:**
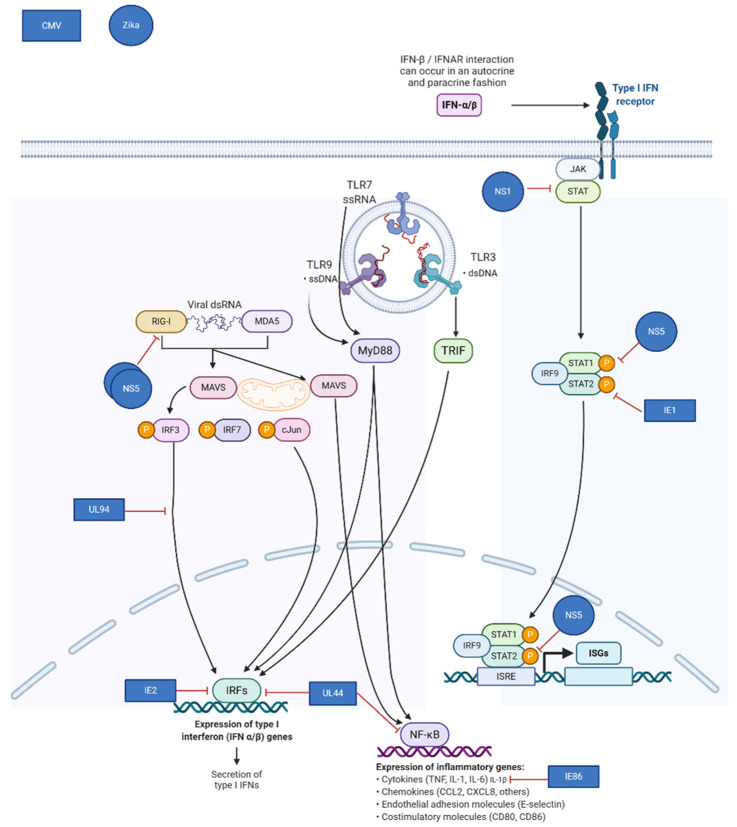
Mechanisms of viral immune evasion in the placenta utilized by cytomegalovirus (CMV) and Zika virus. Adapted from “TLR Signaling Pathway”, by BioRender.com (Accessed on 6 February 2021). Retrieved from https://app.biorender.com/biorender-templates (Accessed on 6 February 2021).

**Figure 3 ijms-22-02921-f003:**
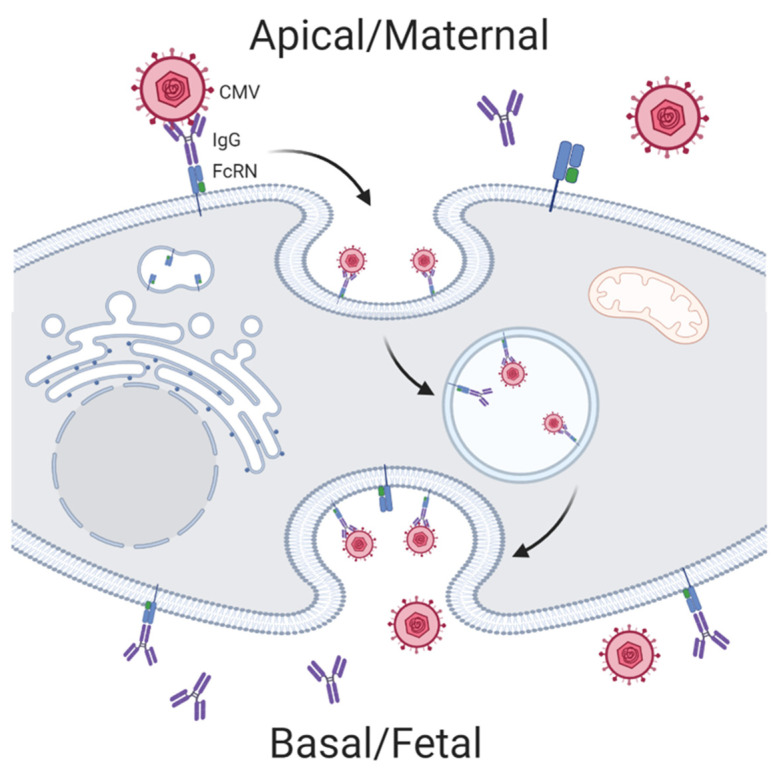
Antibody-dependent enhancement mechanisms hijacked by cytomegalovirus (CMV) for mother to child transmission. Created with BioRender.com.

**Figure 4 ijms-22-02921-f004:**
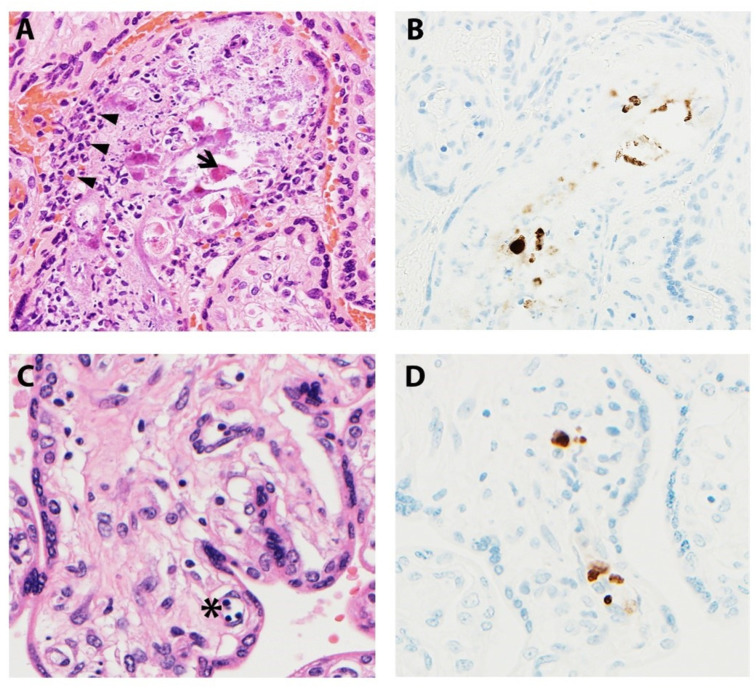
Representative images of placentas infected by CMV. (**A**) Arrowheads illustrate chronic villitis and arrow indicates degraded CMV inclusions and cytopathic effect; (**B**) Immunostaining highlights viral inclusions; (**C**) Asterisk indicates erythroblastosis in early hydropic third trimester placenta; (**D**) Immunostaining highlights scattered positive CMV cells.

**Figure 5 ijms-22-02921-f005:**
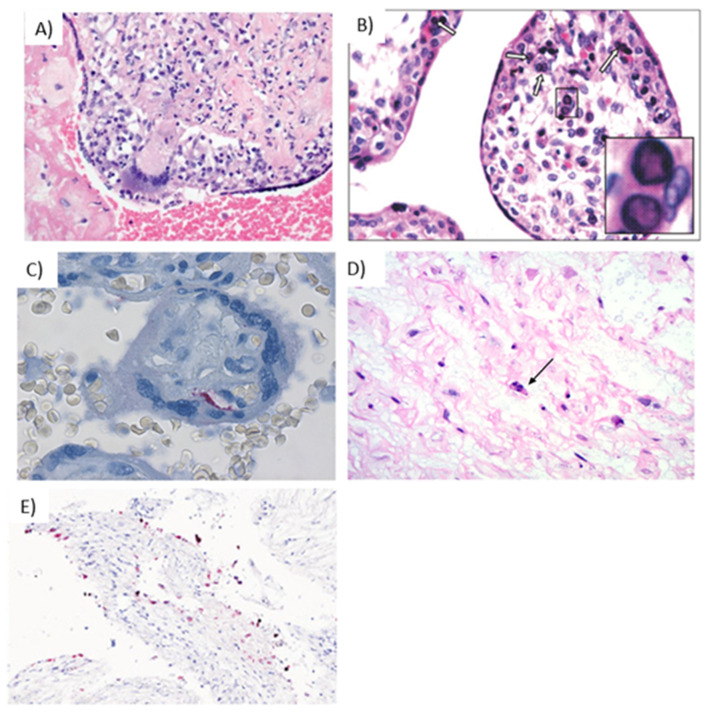
Hallmarks of viral infections in placental tissue. (**A**) Giant cell inflammation caused by varicella infection [142] Reprinted from “Placental and Gestational Pathology” published by Cambridge University Press; reproduced with permission of the Licensor through PLSclear (ref. 47403) (**B**) Parvovirus B19 infection in the trophoblast cells, with intranuclear inclusions (white arrows) in the capillary cells of the chorionic villi, further shown at 40× in the inset [143]. Reprinted from Quemelo *et al.*, 2007; used with permission from SciElo Brazil under the Creative Commons License (CC BY-NC 4.0). (**C**) Rubella infection in the placenta, demonstrated by immunostaining of endothelial cells [126]. Reprinted from *Lazer et al.*, 2016; used with permission from Elsevier (ref. 5010950239116) (**D**) Ascending HSV infection shows infected cells in the cord stroma with nuclear irregularity, hyperchromasia, and individual cell necrosis (arrow) [69]. Reprinted from Heerema-McKenny, 2018; used with permission from John Wiley & Sons, Inc. (ref. 5010940997488) (**E**) Presence of SARS-COV-2 in decidual cells (red) by in situ hybridization (200×) [144]. Reprinted from Menter *et al.*, 2021; used with permission from S. Karger AG.

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
