# Peer review of "Placental Immune Responses to Viruses: Molecular and Histo-Pathologic Perspectives"

_ijms, 2021, doi:10.3390/ijms22062921_

Round 1

Reviewer 1 Report

The authors make a very interesting overview of the placental transmission mechanisms of the main viruses involved in congenital infections. The work is really interesting, well written and readable. Congratulations to the authors for the figures.

I only have small changes to suggest:

Authors. Section Hepatitis Viruses Page 8 lines 297 to 299

Administration of Hepatitis B immunoglobulin (HBIG) can provide an immune barrier by limiting  transcytosis of HBV and reduces the risk of vertical transmission of HBV in pregnancy

Reviewer. Please clarify the time of  HBIG administration in pregnancy  to reach efficacy, recommended doses if possible and change the reference 53 which is marked in superscript and refers to CMV infection.

Authors. Section VZV page 10 lines 362 to 365

The VZV immunohistochemical stain utilizes a mix of antibodies that highlight several glycoproteins, the nucleocapsid protein and the immediate early protein of the virus. As a result, there  is membranous and cytoplasmic staining.

Reviewer. The sentence should be placed in context. The meaning placed in this position is not well understood.

Authors. Section HSV1-2 Viruses Page 13 lines 515 to 519

Neonatal HSV infections demonstrate a myriad (improper term) of manifestations, such as cutaneous (vesicular or ulcerative skin lesions), disseminated disease (sepsis-like picture), and ocular/central nervous system involvement (vision impairment, encephalitis, bulging fontanelle, generalized seizures, lethargy and irritability)

Reviewer. Please change characteristics of HSV 1-2 infection in neonates. “Neonatal HSV infection may present mainly with three clinical manifestations: a) skin, eyes, and mouth herpes (SEM) disease, usually presenting as a vesicular rash occurring at 10–12 days of life……..; b) CNS disease, with (60-70%) or without skin involvement, with clinical manifestations of encephalitis, potentially starting at any time within the first month of life…….; c) disseminated disease, involving multiple organs including CNS, lungs, liver, adrenal, skin, eye, and/or mouth, with about 40% of infants never developing a vesicular rash…….. “

Reviewer 2 Report

The authors of the submitted review 'Placental immune responses to viruses: molecular and histopathologic perspectives' make a thorough case about the importance of the placenta as a barrier protecting the fetus form viral infection. It is a nicely structured review with adequate and  fitting  figures to it.

Moreover, the discussion of both the molecular mechanisms and pathways and the histopathologic hallmarks following infection are sufficiently well described throughout the manuscript. 

The reviewer would expect to see more details regarding the putative molecular queues delivered by viral infection in the case of SARS-CoV-2 infection, and maybe more results from the RNA-seq studies published. . 

Some references that could be added, elucidating the placental pathology and some more deep sequencing results in the case of Covid are 

  1. SARS–CoV-2 infection of the placenta by Hosier et al.
  2. Chronic Histiocytic Intervillositis with Trophoblast Necrosis are Risk Factors Associated with Placental Infection from Coronavirus Disease 2019 (COVID-19) and Intrauterine Maternal-Fetal Severe Acute Respiratory Syndrome Coronavirus 2 (SARS-CoV-2) Transmission in Liveborn and Stillborn Infants

Finally, the manuscript could benefit from a light proofreading, also regarding the spaces in the ‘brackets’ before and after the inserted citations.

Other than the above, the manuscript makes for a very interesting read and should be accepted for publication.
